**Data Availability Statement:** All relevant data are within the manuscript and its Supporting Information files.

# Policy effects of reduced pension contribution rate on innovation input——Empirical evidence from Zhejiang and Shandong provinces

**Zimian He, Jianwei Xu**[ID]*

School of Internet Economics and Business, Fujian University of Technology, Fuzhou, China

* 715057871@qq.com

## Abstract

Reducing pension insurance rates is an important measure for lowering the burden on enterprises. Does this measure encourage enterprises to invest in innovation? This study uses provincial panel data of China during 2000–2017 to analyze the reduction in the proportion of enterprise contributions to pension insurance in Zhejiang and Shandong provinces as a quasi-natural experiment. Specifically, it uses the synthetic control method to construct a counterfactual state of the treatment group, and analyzes the effect of fee reduction policy on enterprise innovation input. Taking R&D intensity as a proxy for innovation input, the study finds that the cost reduction policy has a significant promoting effect on innovation input. Although the reduction of pension insurance rate in Shandong is smaller than that in Zhejiang, the net effect of policy on R&D intensity is higher than that in Zhejiang. This study uses a placebo test, permutation test, and difference-in-differences method to carry out robustness checks, and confirms that the results are reliable, showing that the reduction of pension premium is conducive to the increase in enterprises' R&D intensity. However, owing to different economic development cycles in different regions, the net effect of pension premium reduction on enterprise innovation shows heterogeneity.

## 1. Introduction

Since Solow discovered the role of technological progress in economic growth models, innovation has gained recognition as a key driver of productivity and economic growth. To obtain assistance from technological progress in economic growth, various countries have promoted innovation investment through tax incentives, research and development subsidies, and other policies, aiming to establish an innovative country. At the beginning of this century, China issued a plan to establish an innovative country, attempting to change its economic growth model from factor-driven to innovation-driven. China's R&D expenditure was less than 0.7% at the end of the 20th century; in 2014, the proportion of investment reached 2.02%. Since then, the R&D intensity has steadily increased. In this process, enterprises provide more than 70% of the R&D expenditure funds, and the contribution of enterprises is the greatest.

**Funding:** This work was supported by the Fujian Provincial Social Science [Grant No. FJ2022B161], the start-up funds of Fujian University of Technology [Grant No. GY-S21009] and Fujian Provincial Department of Science and Technology Innovation Strategy Research Project [Grant No.2023R0059]. The contract number that supported my work is now included in 'Funding Information'. The funders had no role in study design, data collection and analysis, decision to publish, or preparation of the manuscript.

**Competing interests:** The authors have declared that no competing interests exist.

Enterprise investment is the backbone of the national innovation system, but in recent years, the difficulties in financing and heavy tax burden of small and medium-sized enterprises have seriously disrupted their investment behavior, which is not conducive to the sustainable development of enterprises. To reduce the burden on enterprises, the country has introduced multiple tax and fee reduction policies, and the reduction of statutory pension insurance rates is an important measure to reduce costs. Is a reduction in rates beneficial for enhancing business vitality? The Netherlands has introduced the Promotion of Research and Development Act (WBSO) to encourage innovation in enterprises, which reduces the wage costs of R&D personnel by reducing payroll taxes and social insurance contributions. Research has shown that this policy helps enterprises increase their R&D [1]. Enterprises' R&D costs will reduce by 10% and R&D intensity will increase by approximately 20% in the short term. Among these, wages and equipment supply account for the majority of the growth in R&D expenditure [2].

From a micro perspective, tax incentives can promote enterprise innovation because reducing taxes can increase after-tax profits and make enterprises more capable of participating in the research and development of new technologies and products. Empirical research on innovation mostly focuses on the enterprise level, but the conclusions of empirical research are divergent (Hall and Rosenberg 2010; Griliches 1984) [3, 4]. A causal relationship exists between tax incentives and corporate innovation [5, 6]. R&D incentives have a positive impact on enterprise innovation [7, 8], R&D incentives have a positive impact on enterprise innovation [9, 10], R&D incentives have a positive impact on enterprise innovation [11]. Tax subsidies may increase the salaries of R&D personnel and have no impact on innovation output [9, 12–14]. Research on OECD countries shows that R&D tax incentives can protect existing enterprises, harm potential entrants, and thus slow down the redistribution process [15]. The quantity of R&D in a country responds to changes in its R&D costs [16, 17]. A similar conclusion was reached when studying the location decisions of American state enterprises: tax incentives increase R&D in that state but reduce R&D in neighboring states, resulting in a net effect of incentives on R&D close to zero [8].

Conclusions regarding the relationship between China's tax incentives and corporate innovation differ. Preferential income tax policies promote enterprise research and development investment, which has a positive incentive effect on enterprise innovation [18–20]. Li and Guo found that tax incentives for high-tech firms only have a significant positive effect on innovation investment and no significant impact on R&D output or industrial development in the short term [21]. Lin et al. considered that there is a significant inverted U-shaped relationship between the intensity of tax incentives and enterprises' innovation level of enterprises [22]. Dai and Liu compared and analyzed the effects of tax incentives and financial subsidies on enterprise innovation, and found that tax incentives have stronger innovation incentive effects [23], while special appropriations have no significant effect [24]. However, Guo considered that innovation subsidies will encourage enterprises to increase R&D investment and substantive innovation output [25].

Does the tax reform's tax reduction effect motivate enterprises to innovate? Howell analyzed the relationship between value-added tax reductions and corporate innovation. In the face of increased financing constraints, private enterprises can stimulate the production of new products and increase sales revenue through transformation tax reductions; however, these have no impact on private enterprises' R&D [26]. Replacing business tax with value-added tax significantly reduces the average increase in tax burden of enterprises and promotes innovation, while innovation output is directly proportional to the decrease in tax burden of enterprises [27]. However, Sun considered that the effect of replacing business tax with value-added tax on enterprise R&D was not significant: the larger the scale, the smaller the R&D efforts of enterprises [28]. The incentive effect of replacing business taxes with value-added

taxes on enterprise technological innovation is more significant for pilot enterprises, non-state-owned enterprises, enterprises with more government subsidies, and enterprises that apply low value-added tax rates [29].

A strong causal relationship exists between tax rates and corporate innovation, and a decrease in effective tax rates benefits the input and output of corporate innovation [30]. While studying the impact of taxation on corporate innovation, Chinese scholars mainly focus on income tax incentives, value-added tax reform, and the policy of replacing business tax with value-added tax. As for whether changes in pension insurance rates affect corporate innovation, they care more about the study of microdata and find a non-linear relationship [31, 32] and a reverse pressure effect [33] between the two aspects. Yet no macrodata has been involved. The pension insurance in this article refers to urban employee pension insurance, and the contribution rate refers to the contribution rate of enterprises, excluding the individual contribution rate. Therefore, the cost reduction policy also refers to the reduction of the enterprise contribution rate of urban employees. As an important component of enterprise labor costs, pension insurance rates can affect enterprises' R&D costs, thereby affecting their investments in R&D. Therefore, this study takes the reduction of enterprise contribution rates for pension insurance in Zhejiang and Shandong provinces as a quasi-natural experiment to conduct research, including the following points. In China, no study utilizing macrodata has been carried out to explore the relationship between pension insurance fee reduction policies and enterprise innovation investment, and this article fills this gap. This study uses the synthetic control method to construct a counterfactual reference group for the processing group, overcome sample selection bias and policy endogeneity issues, and analyze the net effect of cost-reduction policies on enterprise innovation investment.

## 2. Institutional background and theoretical framework

### 2.1 Institutional background

Starting May 1, 2019, the enterprise contribution rate of urban employee pension insurance was reduced. Provinces in which the contribution rate of pension insurance units is higher than 16% can now reduce it to 16%; provinces with less than 16% need to study and propose transitional measures (Guo [2019] No. 13). This is the first policy to significantly adjust the proportion of enterprise contributions nationwide after the implementation of the basic pension insurance system for urban employees, aiming to reduce the burden of labor costs on enterprises and lay the foundation for the national coordination of basic pension insurance. The current urban employee pension insurance system in China began with Document No. 77 issued by the State Council in 1986, which determined that the retirement funds of employees came from the pension insurance accounts jointly paid by enterprises and individuals. The final document determining the payment ratio of both parties was Guo Fa [2005] No. 38. However, owing to the varying levels of economic development in different provinces, there are differences in the payment ratios of enterprises.

Overall planning of the proportion of pension insurance contributions was based on city-level planning, followed by provincial-level planning. Most provinces (cities) implement the guiding rate set by the state, whereas a few provinces (cities) implement a standard different from the national standard because of the unique nature of economic development. Guangdong province, with a large migrant population, is at the forefront of China's reform and opening-up. Before the implementation of a provincial unified planning standard of 14%, there was a significant difference in the payment proportions of pension insurance companies in different cities. In 2002, the Fujian province established an enterprise contribution ratio of 18% for pension insurance. In 2009, Zhejiang province set the proportion of enterprise contributions

to pension insurance at 12%–16%. Previously, the proportion of enterprise contributions to pension insurance in various cities was 20%. In 2012, the province unified its policy by 14%. The 18% contribution ratio of enterprises to pension insurance in Shandong province was stipulated in two separate documents issued in 2010 and 2011. Therefore, the decrease in the proportion of pension insurance contributions in Zhejiang and Shandong provinces provides a quasi-natural experiment to evaluate their net effects on enterprise innovation.

As shown in Table 1, the adjustment of the enterprise contribution ratio of pension insurance in Zhejiang province began in 2008; however, the final provincial pooling rate of 14% was determined in 2009. The 18% contribution ratio of pension insurance companies in Shandong province was gradually adjusted through the policies issued in 2010 and 2011. The adjustment of enterprise payment rates for pension insurance in Zhejiang and Shandong occurred during and after the financial crisis, respectively. Therefore, were changes in pension insurance payment policies in these two provinces affected by the economic crisis? Before the financial crisis, a significant feature of the economic development in Zhejiang was its high dependence on trade. In 2007, the proportion of imports and exports to GDP was 80.65%, which was 17.86% higher than the national average level. In 2008, the financial crisis caused significant fluctuations in the import and export trade of Zhejiang, resulting in a 4.6% year-on-year decline in the GDP growth of Zhejiang that year, which was in line with the national trend. In 2009, both

**Table 1. Content of pension insurance adjustments in Zhejiang and Shandong provinces.**

| Province | File number | Main content |
|---|---|---|
| Zhejiang | Zhe Lao She Lao [2003] No. 59 | Starting from January 1st, 2003, the payment ratio of enterprises was uniformly adjusted to 20%. Enterprises (including collective enterprises) with a payment ratio of less than 20% will pay according to the adjusted payment ratio from now on. |
| Zhejiang | Zhe Zheng Fa [2008] No. 70 | Since 2008, the payment ratio of enterprises has gradually been unified at 12% -16%. |
| Zhejiang | Zhe Zheng [2009] No. 34 | Starting from June 10th, 2009, the provincial payment ratio of enterprises is tentatively set at 14%. If it is higher than this standard, the rate will be reduced. If it is lower than this rate, it can remain unchanged temporarily. In 2012, the province implemented a unified payment ratio. |
| Zhejiang | Zhe Ren She Fa [2009] No. 60 | Starting from July 1st, 2009, for enterprises that pay pension insurance at the provincial social security center, the proportion of pension insurance contributions has been adjusted from 20% to 14%. |
| Shandong | Lu Zheng Ban Fa [2003] No. 37 | All types of urban enterprises and their employees should be included in the scope of urban enterprise employee pension insurance. |
| Shandong | Lu Zheng Fa [2006] No. 92 | Cities with an enterprise contribution ratio of less than 20% will be adjusted to 20% within 3 years. Among them, cities with a corporate contribution ratio of less than 20% and unable to offset the current period's income will be adjusted to 20% in 2006. |
| Shandong | Lu Zheng Fa [2009] No. 108 | The enterprise contribution ratio of pension insurance is 20%,and some cities with a ratio higher or lower than 20% should gradually unify it to 20% according to the principle of overall fund income and expenditure. |
| Shandong | Lu Zheng Fa [2010] No. 61 | Starting from July 1, 2010, the proportion of enterprise contributions to basic pension insurance for enterprise employees in regions with a rate higher than 20% (including 20%) will be reduced by one percentage point. |
| Shandong | Lu Ren She Fa [2011] No. 84 | Districted cities with the proportion of enterprise contributions to pension insurance is higher than 19%(including 19%) and enterprises directly under the provincial management of pension insurance can reduce their contribution ratio by 1%,Jinan City can reduce the payment proportion of enterprises by 2%. |

Note: According to public information on the websites of the People's Governments of Zhejiang and Shandong provinces as well as human resources and social security websites.

import and export trades experienced negative growth, and GDP growth continued to decline by 1.2% annually, with a decline rate of 0.7% higher than the national average. The downward pressure on the economy of Zhejiang continues to increase and Shandong has a high degree of dependence on foreign trade. The financial crisis had a certain impact on import and export trade, but its impact was not as significant as that on Zhejiang. When facing the impact of the financial crisis on the real economy, China has launched a series of economic policies, including the New Social Security Policy, which is mainly to adjust parameters to help enterprises reduce costs. The specific measures of the New Social Security Policy include '5 postponements and 4 reductions', which means implementing a flexible social insurance payment policy. For difficult enterprises affected by the financial crisis and temporarily unable to pay, it is allowed to delay the payment of five social insurance items: elderly care, medical care, unemployment, work-related injury, and childbirth. The delay period is within 2009, and the maximum delay period does not exceed 6 months; Allow difficult enterprises to periodically reduce medical, unemployment, work-related injury, and maternity insurance premiums for a maximum period of 12 months.Therefore, the adjustment of the pension insurance payment ratio in Zhejiang during the financial crisis and in Shandong after the financial crisis were, to some extent, affected by the financial crisis.

## 2.2 Theoretical framework

**2.2.1 Pension insurance and enterprise employment costs.**   Enterprise innovation usually aims at obtaining cost-competitive advantages by extracting higher profit margins. However, how the reduction of pension insurance rates affects enterprise innovation requires clarifying the relationship between pension insurance and enterprise labor costs. (1) The payment of pension insurance by enterprises has increased their labor costs. Pension insurance is an indispensable component of the cost of employment for enterprises. Chinese employee wages are rigid, which makes it difficult for enterprises to fully transfer the burden of pension insurance contributions to employees. Therefore, payment of pension insurance fees increases cost of employment for enterprises [34]. (2) Excessive pension insurance premiums reduce enterprises' enthusiasm and market competitiveness in participating in insurance. The higher the statutory contribution rate of pension insurance, the higher the relevant fees that enterprises hire employees to pay, which increases their motivation to avoid payments. The audit results of Shanghai's 2002–2004 corporate pension insurance payments showed that 80% of enterprises experience this phenomenon [35]. According to the 2018 White Paper on Chinese Enterprise Social Security, only 27% of Chinese enterprises fully comply with social insurance contributions. From the perspective of enterprise operating costs, higher pension insurance rates can have a negative impact on the production and operation activities of enterprises. For example, an increase in statutory welfare fees can lead to high labor costs, prompting enterprises to increase product prices, reduce market share [36], the reduction of non-legal welfare fees, downsize personnel, and lose employment opportunities in low-rate areas [37]. High pension premiums crowd out formal employment. High rates can lead to irregular employment in the labor market, driven by companies' desire to reduce the associated welfare expenditure [38]. In most Latin American countries, owing to the high payroll tax, formal employment is restricted and occupies formal employment space, and reducing the rate is conducive to improving the formal employment rate [39, 40].

*1. Labor cost and enterprise innovation.* The traditional theory of induced innovation posits that technological progress primarily arises from firms' efforts to substitute and conserve expensive factors of production. According to this theory, labor-saving innovations are driven by the relative cost of labor exceeding that of capital [41]. Neoclassical economics, endogenous

growth theory, and related frameworks also support the notion that high wages encourage firms to innovate. In essence, increasing labor costs lead to a reduction in labor inputs, which in turn prompts an increase in capital inputs [42]. A substantial body of empirical research corroborates the view that rising labor costs "push" firms toward innovation [43], and that a phenomenon of "capital-skill complementarity" exists [44]. The theory of induced technological innovation further suggests that rising labor costs, while inducing factor substitution, compel firms to engage in labor-saving technological innovation, thereby leading to biased technological progress [41, 45]. Additionally, higher real wage growth accelerates the process of "creative destruction", as firms that innovate gain a comparative advantage and accrue benefits, including monopoly rents, from their innovations. Consequently, innovative firms are better equipped to navigate an environment of rising labor costs compared to technologically lagging firms, thereby further stimulating innovation [46].

Rising labor costs can enhance enterprise innovation through the effects of human capital. According to the hypotheses proposed under the labor force conversion model and the Stiglitz-Salop separation model, higher wages significantly reduce a firm's labor force turnover, leading to more stable employment relationships, which, in turn, maximize the benefits of early investments [47, 48]. The stability of employment relationships is closely linked to a firm's decisions regarding human capital investments. Typically, firms allocate their R&D investments primarily toward high-tech equipment and R&D staff salaries [49]. Notably, the costs associated with hiring and firing researchers on a temporary basis are substantial. Given that researchers possess specialized knowledge, the processes of recruiting, training, and dismissing specialists can result in significant adjustment costs for R&D activities [50].

*2. Tax |incentives and enterprise innovation.* Pension insurance in China pays the related fees in the form of rates, whereas in foreign countries, it is mostly collected in the form of payroll taxes. Reducing the rates implies implementing preferential policies for payroll taxes, which will reduce the employment costs. Innovation is the main source of modern economic growth. However, because of the externalities of technological progress, private R&D returns are lower than social returns. Therefore, public subsidies are necessary to promote private R&D. Tax incentives are beneficial to enterprise innovation. Tax incentives reduce taxes payable by enterprises, which is beneficial for increasing their after-tax profits, thereby increasing promised returns to shareholders and allowing operators to obtain investments [51]. Innovative projects in low-tax areas are likely to receive financing. Assuming the two regions have different tax rates and similar companies, shareholders are more willing to invest in companies with lower tax rates and higher post-tax profits. Enterprise innovation carries significant risk. Due to adverse selection in the capital market, internal funds are the preferred financing option for innovative projects [52]. Under the same conditions, companies with lower tax rates may have more abundant cash reserves, which are sufficient to provide more financial support for innovation projects. Low-tax areas are more likely to attract R&D personnel, and the employee incentive system depends on post-tax profits, including annual bonuses, long-term incentive plans, and stock options. Assuming that human capital is scarce, talented employees are more willing to join companies with low income tax rates and high post tax profits [53]. The income tax rate affects inventors' international migration. The effective tax rate will be reduced by 1% and the proportion of technical personnel will increase by 14% [30].

The policy of reducing pension insurance premiums as an important measure to reduce enterprise costs reduces the statutory welfare expenses paid by enterprises to employees, thereby reducing their employment costs and improving their profitability. This policy mechanism is similar to enterprises that enjoy tax incentives. Therefore, the theory of tax incentives can be used to analyze the impact of this policy on enterprise innovation. A decrease in the proportion of enterprise contributions to pension insurance results in a corresponding

reduction in the statutory welfare benefits paid by enterprises to employees, improves their operational status, and increases cash reserves. As innovation is a highly uncertain process, companies with more cash reserves will be better able to adapt to adverse outcomes and carry out continuous innovation. The most important investment factor for enterprise innovation is R&D personnel, whose salary level is higher than that of ordinary employees. Statutory benefits comprising wages are inevitably higher, and a decrease in the contribution rate of pension insurance companies will inevitably result in a greater reduction in the statutory benefits of R&D personnel. Enterprises with many R&D personnel can enjoy more benefits from cost reductions, which may have more incentive effects on such enterprises. A decrease in the pension insurance contribution rate does not affect the employee retirement benefits. The decrease in labor costs caused by lower rates is only due to the relative advantage enjoyed by enterprises due to local regulations and policies, rather than the absolute advantage of their product competitiveness. This advantage gradually diminishes with an increase in employee wages. Based on this analysis, a pension insurance fee reduction policy can promote innovative investments by enterprises and improve product competitiveness.

## 3. Research design

### 3.1 Data source and variable descriptions

**3.1.1 Data source.**  This study uses balanced panel data from 31 provinces (cities) in China from 2000 to 2017 to empirically analyze the impact of policies to reduce pension insurance contribution rates on enterprise innovation in the Zhejiang and Shandong provinces. The data mainly came from the *China Science and Technology Statistical Yearbook*, *China Labor Statistical Yearbook*, *China Population and Employment Statistical Yearbook*, and *Statistical Yearbook*. The empirical goal of this study is to use the weighted average of other provinces (cities) to simulate the innovation situation of enterprises in provinces where pension insurance premiums have not been reduced and then compare it with the actual innovation situation of enterprises after the implementation of the fee reduction policy to estimate the net effect of the fee reduction policy on enterprise innovation. Pension insurance rates in Guangdong, Fujian, Shandong, and Zhejiang provinces are not based on national guidance rates as provincial standards. Based on a background analysis of the previous system, Guangdong and Fujian were not suitable research samples. Therefore, this study uses Zhejiang and Shandong as the treatment group and the remaining 27 provinces (cities) as the control group. Starting from May 1, 2016, some provinces (cities) have gradually reduced their endowment insurance contribution rate by 1%, with the employer contribution rate being 19%, which is still higher than the rates in Zhejiang and Shandong provinces. Therefore, this study considered these provinces (cities) as national pooling areas and did not exclude them from the reference group.

**3.1.2 Variable description.**  *Explanatory variables*. The measurement indicators of enterprise innovation mainly include input and output. As macro data cannot separate enterprise patent data, the innovation input indicators used in this paper draw on the research of Hall et al. (2008) [54]. This study uses R&D as the innovation indicator and the intensity of R&D as the proxy variable. In the *China Science and Technology Statistical Yearbook*, there is no relevant data on the internal expenditure of research and experimental development (R&D) of enterprises in various provinces (cities). Total R&D expenditure = enterprise + research and development institution + higher education institution + others. This can be transformed into enterprise + others = total R&D expenditure—research and development institution—higher education institution. Therefore, this study uses the sum of the R&D from enterprises and other departments as an indicator of enterprise innovation investment. The proportion of

R&D from other departments is relatively small, usually less than 2% of the total R&D expenditure.

Policy implementation time setting: Based on institutional background analysis, a corporate contribution rate of 14% for pension insurance in Zhejiang province is stipulated in the 2009 Zhejiang Zhengfa [2009] No. 34 document, whereas the standard for a corporate contribution rate of 18% for pension insurance in Shandong province is stipulated in the Lu Renshe Fa [2011] No. 84 document. Therefore, this study sets the policy implementation times for the two provinces as 2009 and 2011.

*Control variables*. Drawing on existing research, the main factors that affect enterprise innovation investment are set as regional economic development level, investment level, opening-up level, regional population factor, industrial structure, financial level, and financial support. Among them, the level of economic development is measured by per capita GDP and GDP growth rate respectively; The investment level is expressed by the ratio of total fixed assets investment to GDP; The level of opening up to the outside world is measured by the per capita import and export volume; The demographic factors are human capital using the education years of the population aged 6 years and above, population density using the number of people per square kilometer, and population structure using the elderly dependency ratio; The ratio of the output value of the second and third industries to GDP is chosen as the proxy variable for industrial structure; The financial level is the per capita deposit balance of financial institutions at the end of the year; Financial subsidies adopt the ratio of technology expenditure to fiscal budget expenditure.

## 3.2 Evaluation method

The synthetic control method (SCM) is an evaluation method proposed by Abadie and Gardeazabal [55], which can construct a counterfactual reference group of individuals in the treatment group using the weighted average of the control group and simulate the composite value of the dependent variable of a certain policy implementation region in which the policy is not implemented. The difference between the synthetic and true values is the net effect of the policy implementation. The main idea of this study is to consider the regions that have implemented the pension fee reduction policy as the treatment group, and other regions that have not implemented the fee reduction policy as the reference group. The composite control group of the treatment group was constructed by weighing the regions that did not implement the fee reduction policy. The composite control group performed better than the subjectively selected reference group. The counterfactual state of the processing group is constructed according to the data of the synthetic control group, that is, the synthetic value, which needs to have a good fit with the real value before the implementation of the policy. The weight of each region in the reference group in the construction of the counterfactual state is positive, and the sum is equal to 1.

The difference-in-differences (DID) method is the first choice for evaluating policy effects. However, the application of the DID method requires the treatment and control groups to be comparable before policy implementation; that is, the classical DID method needs to meet the assumed conditions of a common trend. However, the samples used in this study hardly meet the abovementioned assumptions, and the synthetic control method can overcome the defects of the DID method. The SCM can estimate the effect of pension insurance fee reduction policy on enterprise innovation more objectively and accurately, while the DID method is subjective and arbitrary in the selection of control group, which is not convincing.

The reduction in pension insurance rates in Zhejiang and Shandong provinces was, to some extent, affected by the financial crisis. Therefore, direct application of the DID method

produces biased estimation results, whereas the SCM is a non-parametric evaluation method that can effectively reduce subjective bias and avoid the problem of policy endogeneity through data-driven weight construction. Therefore, this study chose the SCM as the benchmark evaluation method and the DID method as the test method to test the conclusions.

### 3.3 Statistical description

Fig 1 shows the proportion of enterprise internal R&D expenses to GDP in 31 provinces (cities) in China before and after the implementation of the two provinces' cost-reduction policies.

Although the years of adjustment for the reduction in pension insurance premiums in Zhejiang and Shandong provinces were different, they were both after the 2008 financial crisis, and the intensity of adjustment in Zhejiang province was greater. As there was no change in the innovation investment of the two provinces after the rate change (Fig 1), both provinces increased their R&D after the rate adjustment, gradually narrowing the gap between the regions with the largest R&D. From 2000 to 2008, the R&D intensity of enterprises in Beijing was the highest, reaching 2.43%, while those of Zhejiang and Shandong were 0.89% and 0.88%, respectively, ranking 7th and 8th in the domestic provinces; however, there was a large gap between them and Beijing. From 2011 to 2017, the R&D of domestic enterprises increased, while the R&D of Beijing weakened, and the R&D intensity was only 2.32%, ranking second in the country, followed by Shanghai, with an R&D intensity of 2.33%. Shandong province ranked sixth in R&D intensity in the country, whereas Zhejiang province's ranking remained unchanged. The R&D intensity of the two provinces reached more than 2%, and the gap with the first place is small, indicating that after the implementation of the fee reduction policy, enterprises in the two provinces increased their investment in R&D.

The first row of Table 2 lists the provinces of the processing group and the provinces (municipalities) of the control group are shown in the first column on the left. Taking the R&D intensity of enterprises in Zhejiang province as an example, the synthesized Zhejiang

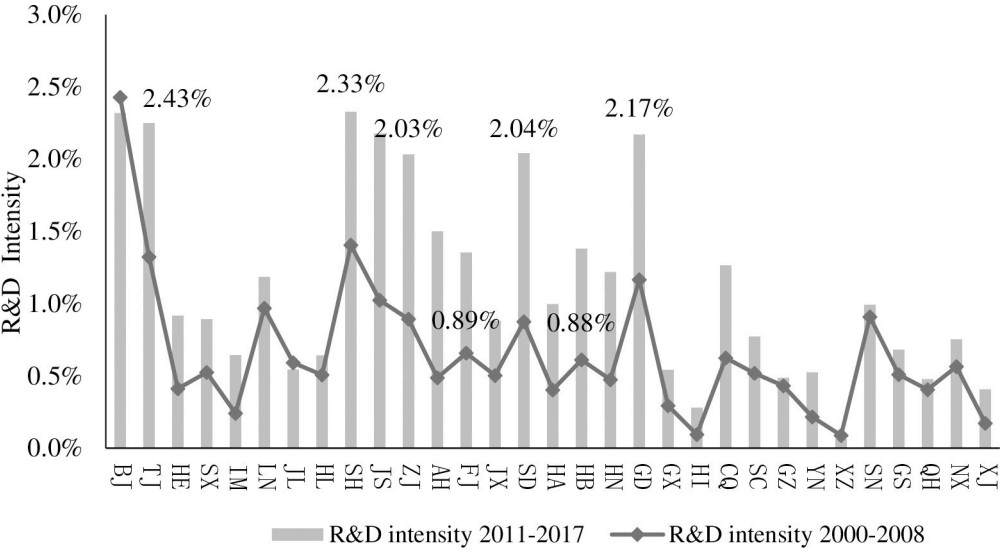

**Fig 1. R&D expenditure intensity before and after the reduction of endowment insurance in 29 provinces(cities).**

**Table 2. Weights of the processing groups in each composite control group.**

| | Zhejiang (ZJ) | Shandong (SD) |
|---|---|---|
| | R&D Intensity | R&D Intensity |
| Beijing (BJ) | 0 | 0 |
| Tianjin (TJ) | 0.018 | 0 |
| Hebei (HE) | 0.007 | 0 |
| Shanxi (SX) | 0 | 0.011 |
| Inner Mongolia (IM) | 0 | 0.014 |
| Liaoning (LN) | 0 | 0 |
| Jilin (JL) | 0 | 0 |
| Heilongjiang (HL) | 0 | 0 |
| Shanghai (SH) | 0.022 | 0 |
| Jiangsu (JS) | 0.831 | 0.739 |
| Anhui (AH) | 0 | 0 |
| Jiangxi (JX) | 0 | 0 |
| Henan (HA) | 0 | 0.145 |
| Hubei (HB) | 0 | 0 |
| Hunan (HN) | 0 | 0 |
| Guangxi (GX) | 0 | 0 |
| Hainan (HI) | 0 | 0 |
| Chongqing (CQ) | 0 | 0 |
| Sichuan (SC) | 0 | 0 |
| Guizhou (GZ) | 0 | 0 |
| Yunnan (YN) | 0 | 0 |
| Tibet (XZ) | 0.02 | 0 |
| Shaanxi (SN) | 0 | 0.03 |
| Gansu (GS) | 0 | 0 |
| Qinhai (QH) | 0 | 0.061 |
| Ningxia (NX) | 0 | 0 |
| Xinjiang (XJ) | 0.102 | 0 |

Note: Abbreviations for the names of each province (city) are in parentheses.

province includes 1.8% of Tianjin city, 0.7% of Hebei province, 2.2% of Shanghai city, 83.1% of Jiangsu province, 2% of Tibet and 10.2% of Xinjiang province, and the weight of each column is 1. When the R&D intensity of enterprises is taken as the dependent variable, the weight of Jiangsu province is the largest in the synthetic Zhejiang province and the synthetic Shandong province, which also indicates that the economic development of Jiangsu, Shandong, and Zhejiang provinces is relatively similar to a certain extent, whereas the weight coefficients of each region in the synthetic Zhejiang and Shandong provinces are different. This synthetic value overcomes the problem of subjective judgment.

# 4. Empirical analysis

## 4.1 Results of the benchmark analysis

The implementation years of the fee reduction policies in Zhejiang and Shandong provinces are different. This study considers 2009 and 2011 as the implementation dates of the policies in the two provinces; when selecting the reference group, provinces with pension insurance enterprise payment rates lower than the national rate are excluded. For example, when

evaluating the impact of fee reduction policies in Zhejiang province on enterprise innovation, Shandong, Guangdong, and Fujian provinces were excluded from the control group, as detailed in Table 2.

**4.1.1 Impact of pension insurance rate reduction on enterprise innovation input in Zhejiang.** The R&D intensity of enterprises in Zhejiang province and synthetic Zhejiang province from 2000 to 2017 is shown in Fig 2A, where the vertical position of the dashed line represents the year in which the pension insurance rate reduction policy was implemented. In Fig 2A, the left side of the dotted line indicates that Zhejiang province and synthetic Zhejiang province have a good fit near the implementation period of the policy. The right side of the dotted line indicates that in the first few years of policy implementation, Zhejiang province and synthetic Zhejiang province still coincide, indicating that this policy has a lag and does not immediately affect enterprises R&D investment. However, around 2013, the trends in Zhejiang province and synthetic Zhejiang province began to diverge, and the gap between the two became increasingly obvious. The R&D intensity of Zhejiang enterprises was higher than that of the synthetic Zhejiang province, indicating that enterprises enjoyed the benefits of a light labor cost burden and gradually increased investment in R&D. The difference between Zhejiang province and synthetic Zhejiang province is the policy effect of pension insurance rate

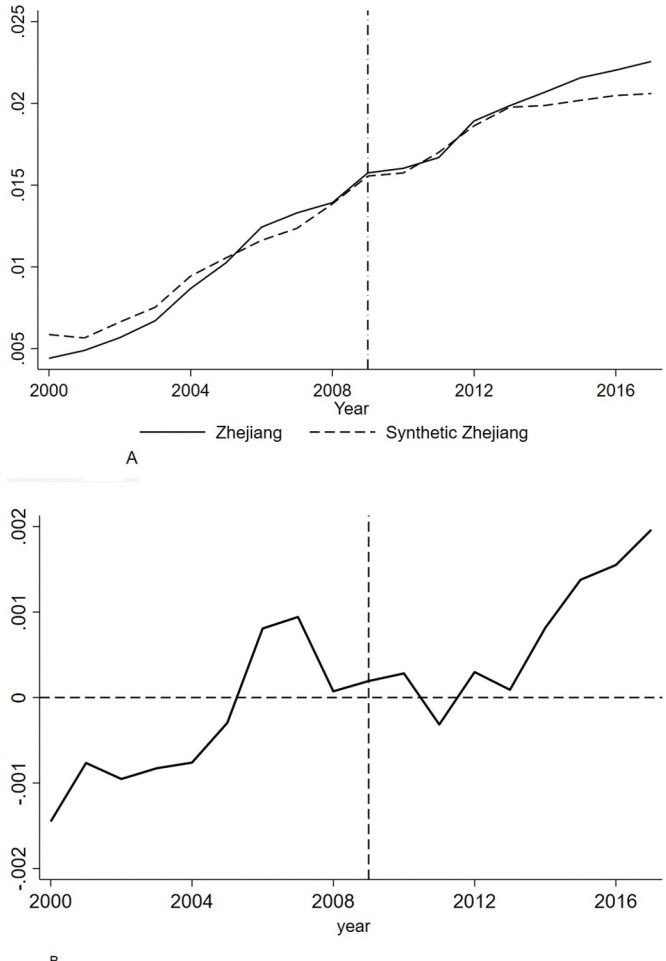

**Fig 2. A.** The True and Synthetic Values of Enterprise R&D Intensity. **B.** The difference between the true and synthetic values of R&D intensity in Zhejiang enterprises.

reduction on enterprise innovation investment, which means that the fee reduction policy can promote the substantial increase of R&D investment of enterprises in Zhejiang province, and this 'promoting efficiency' indicates that the fee reduction policy has significantly improved the R&D intensity of enterprises.

As shown in Fig 2B that during 2000–2009, the gap between the real and synthetic values of the R&D expenditure intensity of Zhejiang enterprises fluctuated by more than 0.1% at the beginning of the period, and the gap fluctuated within 0.1% in the later period; when the pension insurance fee reduction policy was implemented, its innovation effect was not obvious. After 2013, the gap gradually fluctuates, reaching approximately 0.2% in 2017. From 2009 to 2017, according to the real and synthetic values of enterprise innovation, the real average value of R&D intensity of enterprises in Zhejiang province was approximately 1.93%, while the average growth rate of synthetic Zhejiang was 1.86%, which was 0.07% higher than the latter; that is, the fee reduction policy increased the R&D intensity of enterprises by approximately 0.07%.

There is a lag in the net effect of the fee reduction policy on the innovation input of enterprises in Zhejiang province, mainly because foreign trade enterprises were seriously damaged during the 2008 financial crisis, and the main driving force of economic growth in Zhejiang province was import and export trade. In 2008, the financial crisis caused the GDP growth of Zhejiang province to fall by 4.6% annually, which was in line with the decline in the entire country. In 2009, both import and export trades showed negative growth; GDP growth continued to fall by 1.2% year-on-year, the decline rate was 0.7% higher than that of the country, and the downward pressure on the economy of Zhejiang province continued to increase. To reduce the burden of labor costs and invigorate enterprises, the provincial government of Zhejiang province introduced a fee reduction policy during the financial crisis to find a new fulcrum for economic growth. As time went by, the effect of this policy on enterprise innovation input became apparent, and the economic development of Zhejiang province not only got out of the trap of the financial crisis, but also got out of the trap of the financial crisis. Zhejiang has also adopted innovation as a driving force to promote economic transformation and upgrading.

**4.1.2 Impact of pension insurance rate reduction on enterprise innovation input in Shandong.** The R&D intensity of enterprises in Shandong province and synthetic Shandong province from 2000 to 2017 is shown in Fig 3A, where the vertical position of the dashed line represents the year in which the pension insurance rate reduction policy was implemented. In Fig 3A, the solid line represents the true value of the ratio of enterprises' R&D expenditures to GDP, and the dotted line represents the estimated value of the R&D intensity of enterprises, assuming that there is no counterfactual fee reduction policy. Before 2010, the growth paths of Shandong and synthetic Shandong overlapped and synthetic Shandong perfectly replicated the path of enterprise R&D expenditure intensity in Shandong before the implementation of the policy. After 2010, the real value of R&D investment intensity gradually became higher than that of synthetic Shandong, and the gap gradually widened. Although the implementation year of the Shandong policy is set as 2011 in this study, the introduction of this fee reduction policy in Shandong province began in 2010, and the standard for the final implementation of the province's pension insurance enterprise contribution ratio of 18% was 2011. Therefore, in the initial stage of the policy, enterprises' R&D investment had already responded to the policy.

Fig 3B shows the gap between the real and synthetic values of the Shandong enterprises' R&D intensity. From 2000 to 2010, the fluctuation range of R&D intensity gap between Shandong and synthetic Shandong enterprises was about plus or minus 0.1%, and after the reduction of pension insurance enterprise rates in 2011, the R&D expenditure intensity gap increased to 0.3%. According to the true value and synthetic value of enterprise innovation after the implementation of the fee reduction policy, the true value of the R&D intensity of

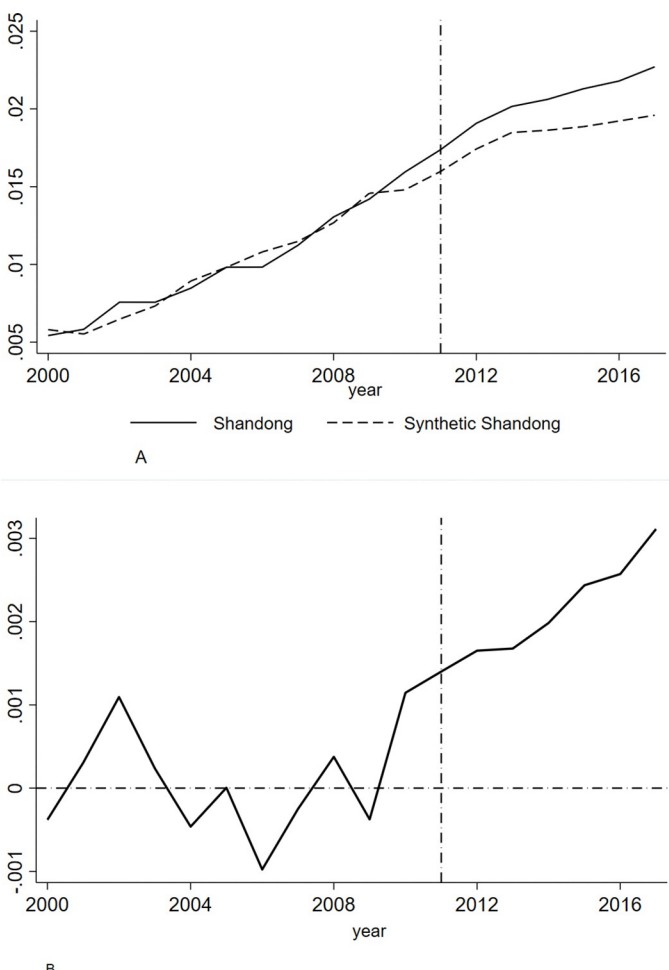

**Fig 3. A.** The Ture and Synthetic Values of Enterprise R&D Intensity. **B.** The difference between the true and Synthetic values of R&D intensity in Shandong enterprises.

enterprises in Shandong is approximately 2.04%, while that of synthetic Shandong is 1.83%; that is, the fee reduction policy has increased the R&D intensity of enterprises by about 0.21%.

In summary, the reduction of pension insurance rate has a significant promoting effect on the intensity of enterprise R&D expenditure, and the 'promoting effect' of Shandong enterprises' R&D investment is greater than that of Zhejiang enterprises. In the year of the implementation of the fee reduction policy, the R&D 'promoting efficiency' in Zhejiang province was not obvious, but after a lag period of 2 to 3 years, this effect began to be more significant, and then the effect gradually increased and was sustainable. In Shandong province, the intensity of enterprise R&D investment 'promoting efficiency' appeared at the beginning of the implementation of the policy, and the effect became increasingly obvious over time. In general, reducing the contribution rate of pension insurance enterprises is conducive to improving the intensity of R&D investments.

## 4.2 Robustness test

**4.2.1 Placebo test.** In this study, the macro data of each province (city) were used as the research sample, and the economic effect of the fee reduction policy on enterprise innovation

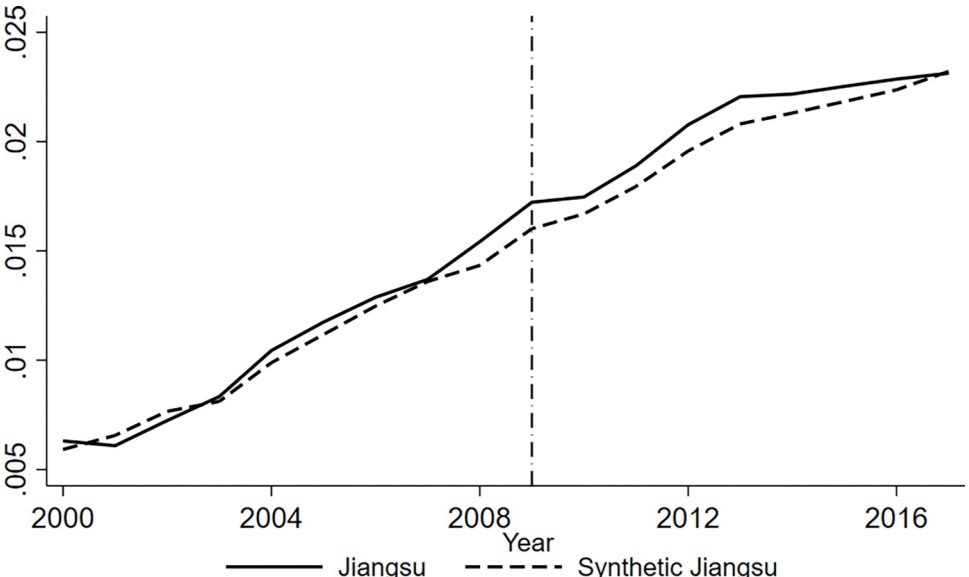

**Fig 4. The true and synthetic values of enterprise R&D intensity.**

input was evaluated using the synthetic control method. Although the subjective judgment problem of the evaluation method is overcome, it is still impossible to determine whether the synthetic value of the treatment group can fit its 'counterfactual' state well, and the estimated results are still uncertain to a certain extent. To test the robustness of the assessment results, the placebo method can be used to test other areas to see if there are other areas that would have the same characteristics as the treatment group.

The specific idea of the placebo test was to select region A from the reference group, assume that the policy intervention was implemented in region A, construct a synthetic control group using the synthetic control method, and evaluate the difference between the synthetic and true values in region A. If the results are similar to those of the treatment group, the synthetic control method does not provide a valid estimate of the net effect of the policy intervention.

A reasonable choice for placebo testing was the region that accounted for most of the weight in the synthetic treatment group. Considering the R&D intensity of enterprises as the explained variable, Jiangsu is the province with the largest weight of synthesis in Zhejiang and Shandong provinces, and the high weight ratio indicates that Jiangsu province is the most similar to Zhejiang and Shandong provinces. The placebo test was adopted in 2009 as the implementation date for the policies, as shown in Fig 4.

Fig 4 shows the placebo test results of the research on the R&D intensity of enterprises with Jiangsu province as the treatment group, indicating that the gap between Jiangsu province and synthetic Jiangsu province did not change significantly before and after the implementation of the policy. This proves, to a certain extent, that the pension insurance fee reduction policy affects the R&D intensity of enterprises, rather than other common accidental factors.

**4.2.2 Permutation test.** To verify whether the 'promoting effect' of enterprise R&D intensity is brought about by the pension insurance fee reduction policy, rather than other unseen external factors, and to evaluate whether the policy effect is statistically significant, this study adopts the permutation test [56] to determine whether there are other regions with the same situation as Zhejiang and Shandong provinces. The basic idea of this method is to randomly select region B in the control group, assume that the policy intervention is implemented in region B in a certain year, use the synthetic control method to construct the synthetic value of

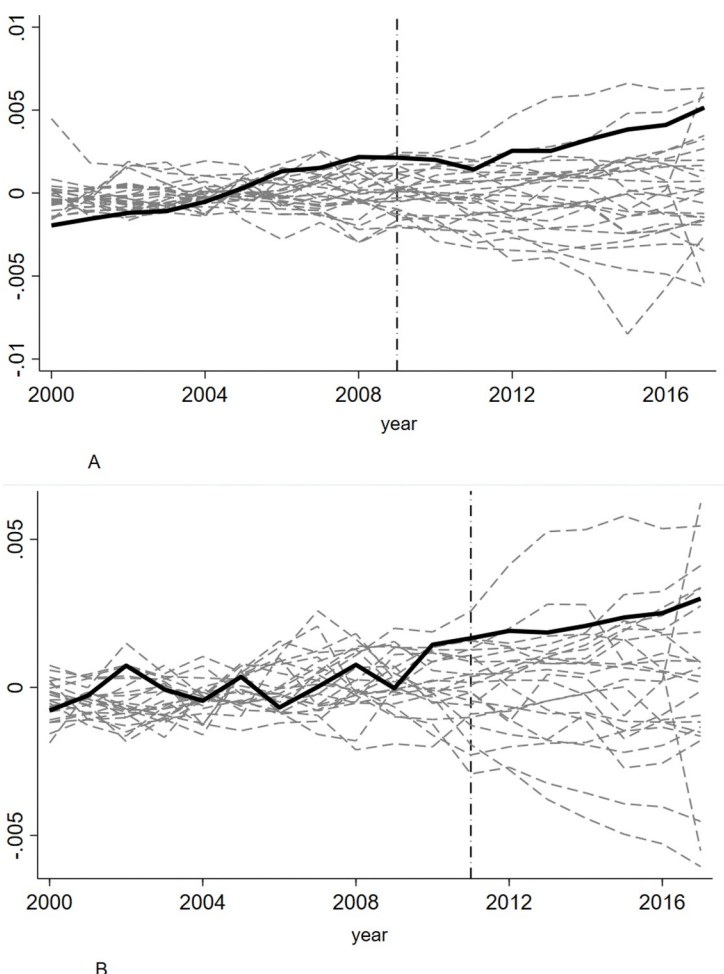

**Fig 5. A.** Permutation Test of R&D Intensity for Enterprises in Zhejiang Province. **B.** Permutation test of R&D intensity in enterprises in Shandong Province.

the counterfactual state, mark the gap between it and the real value as D, and compare D with the difference between the real value and the synthetic value of the treatment group. If the difference is obvious, it indicates that the net effect of the policy intervention is significant because the synthetic control samples of the region before policy implementation have good fitting effects. If the synthetic effect of a region before policy implementation is not ideal, that is, the mean square prediction error (MSPE) value is relatively large, even the larger difference in the predicted variables obtained in the later stage is not sufficient to reflect the implementation effect of the policy. Therefore, when constructing the permutation test, regions with MSPE values greater than twice those of Zhejiang and Shandong before policy implementation were excluded from the 27 control samples. Fig 5A and 5B show the distribution of differences after excluding these regions.

As shown in Fig 5A, before 2009, the gap between Zhejiang province and synthetic Zhejiang province in terms of enterprise R&D intensity was not obvious compared with other regions; however, after 2009, the gap began to gradually increase, indicating that the pension insurance fee reduction policy had a certain promoting effect on enterprise R&D. It also indicates that there is only a 1/27, or 3.70% probability, gap between the R&D expenditure intensity of enterprises in Zhejiang and synthetic Zhejiang, which is similar to the significance level of the

**Table 3.  Influence of the pension insurance fee reduction policy on enterprise innovation input.**

|  | Zhejiang | | Shandong | |
|---|---|---|---|---|
|  | R&D Intensity (1) | R&D Intensity (2) | R&D Intensity (3) | R&D Intensity (4) |
| The net effect of the policy | 0.0013** (0.0006) | 0.0013* (0.0008) | 0.0019 (0.0012) | 0.0029*** (0.0006) |
| Control variables | No | Yes | No | Yes |
| Time effect | Yes | Yes | Yes | Yes |
| $R^2$ | 0.9444 | 0.9872 | 0.7622 | 0.9919 |
| Sample size | 72 | 72 | 72 | 72 |

Note: Robust standard errors are in parentheses

***, **, and * represent significance levels of 0.01, 0.05, and 0.1, respectively.

statistical indicators. Therefore, it can be considered that the increase in R&D expenditure intensity of enterprises in Zhejiang province at the 5% level is significant, which can prove that the increase in R&D expenditure intensity of enterprises in Shandong province is statistically significant.

**4.2.3 DID method.**    To test the effectiveness of the SCM estimation results, the DID method can be used to evaluate the net effect of pension insurance cost-reduction policies on enterprise innovation investment because the processing and control groups must meet the common trend assumption. Therefore, when selecting the control group, this study attempts to select regions that are as consistent with the trend of innovation investment indicators in Zhejiang and Shandong as possible. It was found that the sample size of the control group changed with regional variations. Zhejiang province is the treatment group, and its control group includes Tianjin city, Shanghai city, and Jiangsu province. Shandong province is the treatment group, and its control group includes Jiangsu, Anhui, and Hunan provinces. The empirical results are presented in Table 3.

The results listed in Table 3 show that the net effect of the pension insurance fee reduction policy on enterprises' R&D intensity in Zhejiang and Shandong is significantly positive. Although the fee reduction effort is small in Shandong province, the net effect of innovation input generated by the policy is stronger than that in Zhejiang province, which is consistent with the results obtained by the SCM.

These three robustness tests show that the pension insurance fee reduction policy has an economic effect on enterprises' R&D intensity. Compared with its potential growth trend, the increase rate has been increased to a certain extent, indicating that this fee reduction policy has a 'promoting effect' on the R&D of enterprises and is statistically significant, and the benchmark analysis in this study is reliable.

## 5. Conclusion and discussion

In this study, the reduction of enterprise contribution rates of pension insurance in Zhejiang and Shandong provinces is taken as a quasi-natural experiment, and provincial (municipal) balance panel data from 2000 to 2017 are selected. In this study, the weighted average of the reference group is adopted by the synthetic control method, and the synthetic value of the 'counterfactual' state of the treatment group is constructed. The net effect of the pension insurance fee reduction policy on enterprise innovation input is evaluated by the difference between the true value and the synthetic value. The results show that taking the proportion of R&D to GDP as the proxy variable of innovation input, fee reduction policy has a significant effect on innovation input. This effect has a lag period of two to three years in Zhejiang province, but no lag effect in Shandong province. Although the pension fee reduction in Shandong province

was less than that in Zhejiang province, the net effect of the policy on enterprises' R&D intensity was higher. The placebo test, permutation test, and DID method were used to test the robustness of the results, and the benchmark evaluation results in this study are reliable.

Overall, this paper confirms that the pension contribution rate reduction policy can incentivize firms to increase their R&D investment intensity. However, it also suggests that the effects of this policy on R&D investments is not directly proportional to the extent of the rate reduction; larger reductions do not necessarily lead to more pronounced increases in R&D activity. The national policy of "tax reduction and fee reduction", which includes lowering the statutory pension contribution rates for firms, can encourage them to boost their R&D investments and improve their cost-effectiveness, thereby enhancing the competitiveness of Chinese enterprises in the market. Additionally, the paper highlights that differences in economic development across different provinces result in varying net effects of the pension contribution rate reduction policy on firms' innovation investments. The lagging effect of Zhejiang province's fee reduction policy on enterprise innovation investment also indicates to some extent that the impact of the financial crisis on enterprises in the province is higher than that of Shandong province. From 2008 to 2014, the economic development rate of Zhejiang was slower than that of Shandong, whereas in 2007, the economic growth rate of Zhejiang was higher than that of Shandong. This is mainly because, although both Zhejiang and Shandong are strong foreign trade-oriented economic provinces, Zhejiang province has a higher dependence on imports and exports. The 2008 financial crisis had a greater impact on Zhejiang province enterprises, and it took them time to overcome their difficulties of the financial crisis. Therefore, there is a lag in the net effect of cost-reduction policies on enterprises' innovation investment.

Around 2013, the policy effectiveness of reducing the statutory pension insurance premium rate in Zhejiang province on enterprise innovation investment was evident to some extent because of its economic transformation. Before the financial crisis, the economic development of Zhejiang province relied primarily on imports and exports. After the financial crisis, economic development tended to be investment and consumption, and innovation became the driving force for Zhejiang's economic development The contribution rate of pension insurance enterprises in Zhejiang province was higher than that in Shandong province, which can reduce labor cost expenditures at the enterprise level. Therefore, a lower pension insurance payment rate is conducive to the development of small and medium-sized enterprises in Zhejiang province.

## Supporting information

**S1 Data. Relevant research data.**
(XLSX)

**S1 Appendix. Explanation of synthetic control method.**
(DOCX)

## Author Contributions

**Methodology:** Zimian He.

**Writing – original draft:** Zimian He.

**Writing – review & editing:** Jianwei Xu.

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
