## [Decision Letter · Decision Letter 0]

12 Aug 2024

PONE-D-24-21206Policy Effects of Reduced Pension Contribution Rate on Innovation Input —— Empirical Evidence from Zhejiang and Shandong ProvincesPLOS ONE

Dear Dr. Xu,

Thank you for submitting your manuscript to PLOS ONE. After careful consideration, we feel that it has merit but does not fully meet PLOS ONE’s publication criteria as it currently stands. Therefore, we invite you to submit a revised version of the manuscript that addresses the points raised during the review process.

We look forward to receiving your revised manuscript.

Best regards,

Prof. (Assoc.) Donato Morea, Ph.D.

Academic Editor

PLOS ONE

Reviewers' comments:

Reviewer's Responses to Questions

**Comments to the Author**

1. Is the manuscript technically sound, and do the data support the conclusions?

Reviewer #1: Yes

Reviewer #2: Partly

2. Has the statistical analysis been performed appropriately and rigorously? 

Reviewer #1: Yes

Reviewer #2: Yes

3. Have the authors made all data underlying the findings in their manuscript fully available?

Reviewer #1: Yes

Reviewer #2: Yes

4. Is the manuscript presented in an intelligible fashion and written in standard English?

Reviewer #1: Yes

Reviewer #2: No

5. Review Comments to the Author

Reviewer #1: (1) Why select Zhejiang and Shandong Provinces as research objectives

(2) Please provide literature references for explaining why taking R&D intensity as a proxy for innovation input

(3) Research gaps among prior works should be presented

(4) Theoretical implications should be provided

Reviewer #2: The paper investigates the impact of reduced pension insurance contribution rates on enterprise innovation in Zhejiang and Shandong provinces in China, using data from 2000 to 2017. The final results indicate that lowering pension insurance contribution rates significantly positively affect enterprise innovation in these provinces. This policy measure increases R&D intensity by alleviating financial constraints and enhancing firm profitability, though the effects vary by region and over time.

The empirical strategy is based on provincial panel data from 31 Chinese provinces and employs the synthetic control method (SCM) to construct a counterfactual scenario. This approach helps estimate the impact of reduced pension contributions on R&D intensity. To ensure robustness, the authors also use placebo tests, permutation tests, and the Difference-in-Differences (DID) method.

In Zhejiang province, the reduction in pension contribution rates led to a significant increase in R&D intensity, although this effect exhibited a lag of about 2-3 years post-implementation. The average R&D intensity increased by approximately 0.07% compared to the synthetic control. The impact was more immediate in Shandong province, with the R&D intensity increasing by about 0.21% after the policy implementation. Despite a smaller reduction in pension contributions compared to Zhejiang, Shandong showed a stronger response.

The study makes important contributions to the existing literature. However, before publication, some important limitations need to be addressed. These include its focus on only two provinces, which may limit the generalizability of the findings, and the potential trade-off between reduced mechanization and increased R&D investment that should be further explored.

Why should reducing labor costs through a cut in pension insurance contribution rates incentivize more R&D investments? Reducing labor costs through a cut in pension insurance contribution rates can incentivize more R&D investments for several reasons rooted in economic theory and empirical observations: enhanced liquidity, reduced financial constraints, increased profitability, reinvestment of profits, cost-competitive innovations, strategic shifts, and a buffer against uncertainty. The paper needs to discuss all these aspects to unveil the potential mechanism behind the final results.

The paper does not sufficiently address the long-term sustainability of reduced pension contributions. What are the potential risks or downsides of this policy in terms of pension fund solvency or social security? Policymakers must balance the short-term economic benefits with the long-term sustainability of pension funds. To explain the positive effects of pension insurance cuts, the authors should discuss complementary measures in more detail, such as improving the overall business environment and providing targeted R&D subsidies. An aging population increases the fiscal pressure on public pension systems, and as the number of retirees grows, the demand for pension benefits rises, potentially leading to higher deficits if the contribution base is simultaneously reduced. This demographic shift necessitates reforms that ensure the pension system's sustainability without compromising economic growth. To offset the reduced contributions, governments may need to explore alternative funding mechanisms for pension systems. This could include increased general taxation, introducing or expanding private pension schemes, or implementing policies encouraging higher labor force participation among older workers.

The sample size of 72 units is quite small, and it is unclear how it was achieved, given that the entire analysis is based on provincial panel data from 31 provinces (cities) in China from 2000 to 2017.

I suggest discussing in more detail the assumptions behind the exogeneity of the explanatory variables adopted to control for regional economic development, investment level, openness, population factors, industrial structure, financial level, and financial support with respect to the dependent variable used to capture the R&D intensity.

Although the study aligns with the broader literature demonstrating that innovation is a key driver of economic growth, as highlighted by Griliches (1984) and Hall & Rosenberg (2010), I suggest including additional relevant literature that was not covered in the initial version of the paper, which could provide a broader context or deeper insights into the relationship between fiscal policies, labor costs, and innovation. Some examples are Aghion, P., & Howitt, P. (1992), Romer, P. M. (1990), Klette, T. J., & Kortum, S. (2004), Zwick, E., & Mahon, J. (2017), Bloom, N., Sadun, R., & Van Reenen, J. (2012), Garicano, L., Lelarge, C., & Van Reenen, J. (2016) and Autor, D. H., Dorn, D., & Hanson, G. H. (2013). Incorporating some of these additional references may provide a more comprehensive understanding of the interplay between fiscal policies, labor costs, and innovation. They offer theoretical frameworks, empirical evidence, and methodological advancements that can deepen the analysis presented in the paper. As a minor point, references should be reorganized in alphabetical order.

Finally, I have included a list of additional areas where the paper could be improved before finalizing it for publication:

1. Time Frame: The study uses data from 2000 to 2017, which provides a solid historical perspective but may not capture the most recent trends or long-term effects of the policy changes.

2. Geographical Focus: The analysis is limited to Zhejiang and Shandong provinces. While these are significant regions, the study could benefit from including more provinces to provide a broader view of the policy's impact across China.

3. Regional Differences: The paper acknowledges heterogeneity in economic development but could delve deeper into how specific regional characteristics (e.g., industrial structure and enterprise size) affect policy outcomes. This would help understand why the policy impact differs between Zhejiang and Shandong.

4. Mechanistic Explanation: The study shows that reduced pension contribution rates lead to increased R&D intensity, but it lacks a detailed discussion of the causal mechanisms. How exactly do reduced costs translate into higher innovation investment? Are there intermediate steps or specific areas within R&D that benefit more?

5. Sectoral Analysis: A more detailed breakdown of how different sectors respond to the policy change could provide insights into sector-specific effects and guide more targeted policy recommendations.

6. Theoretical Framework: The theoretical framework could be expanded to include more detailed discussions of related economic theories, particularly those addressing the relationship between labor costs and innovation.

7. Complexity of Methodology: The synthetic control method and other statistical techniques used are complex and might not be fully accessible to all readers. Simplifying the explanation or providing a more detailed methodological appendix could help.

8. Data Presentation: The paper includes various figures and tables, but their presentation could be clarified. More intuitive visualizations or summaries could make the results easier to interpret.

6. PLOS authors have the option to publish the peer review history of their article (what does this mean?). If published, this will include your full peer review and any attached files.

Reviewer #1: **Yes: **YANG Z

Reviewer #2: **Yes: **Francesco Porcelli

---

## [Author Response · Author response to Decision Letter 0]

10 Sep 2024

Dear Editor and Reviewers,

Thank you for your suggestions, which are crucial for further optimization of my paper.

The revisions I have made are marked in blue in this paper.

The revised paper consists of the following parts:

1. A response letter to review comments, which has been uploaded as a separate file and labeled “Response to Reviewers”.

2. The revised paper, titled “Revised Manuscript with Track Changes”.

3. The revised paper without tracked changes, named “Manuscript”.

4. The project funds financing the research in this paper are added to the “Cover Letter”.

Revisions made as suggested:

1. The format of this paper has been adjusted and file of the paper has been named according to the requirements of the journal.

2. Relevant research data have been submitted upon request.

3.The responses to review comments are as follows:

Reviewer 1:

1. Reasons for choosing Zhejiang Province and Shandong Province as research targets.

The statutory pension contribution rate for urban workers in China is first coordinated at the municipal level and then at the provincial level. In practice, most provinces implement the statutory contribution rate of 20%, as mandated by the state. However, a few provinces have adopted different contribution rates due to their unique economic circumstances. These provinces include Guangdong, Fujian, Zhejiang, and Shandong. In Guangdong Province, pension contribution rates varied significantly across cities before the provincial statutory contribution rate of 14% was implemented. Similarly, Fujian Province set its pension contribution rate at 18% as early as 2002. Given these, neither province is suitable for inclusion in this study. In contrast, the pension contribution rates in Zhejiang Province were reduced to a range of 12% to 16% in 2009, while Shandong Province officially lowered its rate to 18% between 2010 and 2011. Prior to these reductions, the employer pension contribution rate in both provinces had been set at 20%. Thus, the lowered pension contribution rates in Zhejiang and Shandong Provinces provide a “quasi-natural experiment” to evaluate the economic effects of the pension contribution rate reduction policy on enterprise innovation.

2. References to explain why R&D intensity is used as a proxy for innovation investments.

A reference has been added to this paper, reflecting that it draws on Hall et al. (2008) to use R&D intensity as a proxy for innovation investments.

3. Research gaps between this paper and previous studies.

Previous research findings have been added to the introduction section along with the differences between the research in this paper and the previous research.

4. Theoretical insights

Theoretical insights are marked in blue in this paper.

Reviewer 2:

Thank you very much for your suggestions about the weaknesses of this paper. The following changes have been made accordingly.

1. Time frame: The selected time period of this paper is 2000-2017, which is mainly because, in 2019, the Chinese government reduced the pension contribution rate from 20% to 16%. At the end of the same year, the COVID-19 pandemic broke out, and over the subsequent three years, the central government temporarily exempted enterprises facing operational difficulties from pension contributions. As a result, this later period is not suitable for the analysis in this study. Therefore, the time frame used in this paper concludes in 2017, excluding the years that followed.

2. Regional selection: Why Zhejiang Province and Shandong Province are chosen as research targets

Following the implementation of the 20% pension contribution rate in China, only four provinces adopted a lower rate: Guangdong, Fujian, Zhejiang, and Shandong. In Guangdong, actual contribution rates varied significantly among different cities, making it less suitable for consistent analysis. Fujian lowered its statutory pension contribution rate to 18% in 2002, which is too distant from the relevant timeframe of this study. On the other hand, Zhejiang and Shandong announced pension contribution rate reductions in 2009 and 2010, respectively, after previously adhering to the national rate of 20%. This provides a “quasi-natural experiment” for assessing the effects of pension contribution reductions on enterprises’ innovation investments, making these two provinces ideal research targets.

3. Regional differences: Addressing the reviewer’s suggestion on industrial structure and enterprise size.

The reviewer’s suggestion to explore specific regional characteristics, such as industrial structure and enterprise size, is valuable. While this paper utilizes macroeconomic data, China’s statistical yearbooks do not provide detailed categorizations of these factors, making it challenging to conduct a more in-depth analysis with the available data. However, recognizing the importance of these dimensions, I intend to gather more comprehensive data in the future to conduct follow-up research.

4. Lack of mechanism analysis

The reviewer’s suggestion is indeed valuable. A detailed analysis of the causal mechanism behind how lowered pension contribution rates impact enterprises’ innovation investments is not fully explored in this paper, which represents a limitation. Existing research suggests that the effects of pension contribution rate reductions on innovation inputs can be examined from multiple perspectives, such as labor costs, financing costs, and tax incentives. However, this paper only briefly touches upon these mechanisms from a theoretical standpoint and lacks the micro-level data needed to further substantiate the conclusion. Since the paper relies on macroeconomic data rather than micro data from individual enterprises—such as employee wages, pension premiums, and financing costs—there is no detailed mechanism analysis included. In future research, I aim to gather micro-level data to provide a more thorough exploration of these mechanisms.

5. Sectoral analysis

The use of macro data limits the ability to conduct sectoral analysis in this paper. This is primarily due to the fact that sector-specific data in China’s provincial statistical yearbooks are neither comprehensive enough nor consistently available over time. Unlike micro data, which include a variety of enterprise-level elements, macro data do not offer the granularity necessary for analyzing different sectors effectively. To address this shortcoming, I plan to carry out further research in this field using micro-level data.

6. Theoretical framework

The theoretical framework of this paper has been extended to provide a more detailed discussion of relevant economic theories, particularly those that address the relationship between labor costs and innovation.

Key theories on labor costs and enterprise innovation have now been incorporated into the theory section.

7. Methodological complexity: Explanation of synthetic control method

An explanation of the synthetic control method has been added to the appendix.

8. Data presentation

The reviewer’s suggestion for clearer data presentation is important. Although this paper includes various graphs and charts, their clarity and intuitiveness could be improved to enhance the reader’s understanding.

The diagrams in this paper, except for Figure 1, are the output from STATA, which makes it challenging to create editable diagrams for inclusion in a Word document, as required by PLOS ONE. According to the journal’s formatting guidelines, diagrams must be submitted as editable figures, which could affect the quality of visualization, especially when figures like Fig. 2(a) and Fig. 2(b) are ideally displayed side by side to intuitively show the R&D expenditure gap between Zhejiang and synthetic Zhejiang. Presenting them in separate columns within a Word document might compromise the visual clarity. This explanation is provided to clarify the potential limitations in achieving optimal presentation according to the specific requirements.

9. References ordering

The references in this paper are arranged based on the order of citation. If required, I can adjust the references to alphabetical order based on the authors’ last names to comply with journal-specific requirements.

---

## [Decision Letter · Decision Letter 1]

3 Dec 2024

Policy Effects of Reduced Pension Contribution Rate on Innovation Input —— Empirical Evidence from Zhejiang and Shandong Provinces

PONE-D-24-21206R1

Dear Dr. Xu,

We’re pleased to inform you that your manuscript has been judged scientifically suitable for publication and will be formally accepted for publication once it meets all outstanding technical requirements.

Best regards,

Prof. Donato Morea

Academic Editor

PLOS ONE

Reviewers' comments:

Reviewer's Responses to Questions

**Comments to the Author**

1. If the authors have adequately addressed your comments raised in a previous round of review and you feel that this manuscript is now acceptable for publication, you may indicate that here to bypass the “Comments to the Author” section, enter your conflict of interest statement in the “Confidential to Editor” section, and submit your "Accept" recommendation.

Reviewer #1: All comments have been addressed

Reviewer #2: All comments have been addressed

2. Is the manuscript technically sound, and do the data support the conclusions?

Reviewer #1: Yes

Reviewer #2: Yes

3. Has the statistical analysis been performed appropriately and rigorously? 

Reviewer #1: Yes

Reviewer #2: Yes

4. Have the authors made all data underlying the findings in their manuscript fully available?

Reviewer #1: Yes

Reviewer #2: Yes

5. Is the manuscript presented in an intelligible fashion and written in standard English?

Reviewer #1: Yes

Reviewer #2: Yes

6. Review Comments to the Author

Reviewer #1: (No Response)

Reviewer #2: After conducting a thorough examination of the revised version of the manuscript, I am pleased to acknowledge the substantial improvements made by the authors in addressing the majority of the requested changes. While some revisions may not perfectly align with the specifics of my initial suggestions, I recognize that the authors have thoughtfully incorporated these adjustments in ways that enhance the overall quality and coherence of the paper. The revisions demonstrate a commendable effort to refine the arguments, improve the structure, and ensure clarity in the presentation of the research findings.

7. PLOS authors have the option to publish the peer review history of their article (what does this mean?). If published, this will include your full peer review and any attached files.

Reviewer #1: **Yes: **YANG ZHEN

Reviewer #2: **Yes: **Francesco Porcelli

---

## [Editor Report · Acceptance letter]

9 Dec 2024

PONE-D-24-21206R1 

PLOS ONE

Dear Dr. Xu, 

I'm pleased to inform you that your manuscript has been deemed suitable for publication in PLOS ONE. Congratulations! Your manuscript is now being handed over to our production team.

Kind regards, 

on behalf of

Professor (Associate) Donato Morea 

Academic Editor

PLOS ONE